# Thermochemical Study of 1-Methylhydantoin

**DOI:** 10.3390/molecules27020556

**Published:** 2022-01-16

**Authors:** Juan M. Ledo, Henoc Flores, Fernando Ramos, Elsa A. Camarillo

**Affiliations:** 1Complejo Regional Mixteca, Campus Izúcar de Matamoros, Benemérita Universidad Autónoma de Puebla. Carr, Atlixco-Izúcar de Matamoros 141, San Martín Alchichica, Izúcar de Matamoros C.P., 74570 Puebla, Mexico; juan.ledo@correo.buap.mx; 2Facultad de Ciencias Químicas, Benemérita Universidad Autónoma de Puebla, 14 sur y Avenida San Claudio, Puebla C.P., 72570 Puebla, Mexico; elsa.camarillo@correo.buap.mx

**Keywords:** energy of combustion, enthalpy of formation, hydantoins, enthalpy of sublimation

## Abstract

Using static bomb combustion calorimetry, the combustion energy of 1-methylhydantoin was obtained, from which the standard molar enthalpy of formation of the crystalline phase at *T* = 298.15 K of the compound studied was calculated. Through thermogravimetry, mass loss rates were measured as a function of temperature, from which the enthalpy of vaporization was calculated. Additionally, some properties of fusion were determined by differential scanning calorimetry, such as enthalpy and temperature. Adding the enthalpy of fusion to the enthalpy of vaporization, the enthalpy of sublimation of the compound was obtained at *T* = 298.15 K. By combining the enthalpy of formation of the compound in crystalline phase with its enthalpy of sublimation, the respective standard molar enthalpy of formation in the gas phase was calculated. On the other hand, the results obtained in the present work were compared with those of other derivatives of hydantoin, with which the effect of the change of some substituents in the base heterocyclic ring was evaluated.

## 1. Introduction

Heterocyclic chemistry is one of the vastest and most complex branches of organic chemistry. It is of great interest due to its theoretical implications, for the diversity of its synthetic procedures, and for the biological and industrial importance of heterocyclic compounds. As a special case, heterocyclic chemical structures that contain nitrogen are very important molecules that participate, from biological processes constituting the nitrogenous bases in RNA and DNA, even in different industries such as pharmaceuticals.

Hydantoins and their derivatives are five-membered cyclic ureas with substituents at different positions on the ring. The structure of these compounds is found in a wide range of biologically active compounds [1] with important pharmacological applications, such as antitumor and anticancer [2,3,4,5,6]; antiarrhythmic and anticonvulsant agents [7,8]; anti-inflammatory and antiviral drugs [9,10]; and recently, they have been used as anti-inflammatory drugs in the treatment of rheumatoid arthritis [11]. Some hydantoin derivatives also have some applications in the agrochemical area as bactericides, fungicides, and herbicides [12,13,14,15].

On the other hand, molecular thermochemistry allows us to have an approach to understand the thermodynamic stability of molecules, and rests on their chemical bonds and the interactions present in them. This knowledge is important since it allows us to establish relationships between energy, structure, and reactivity in molecules. In addition, this information provided by thermochemistry is in high demand in some areas of science and technology.

In the above context, a thermochemical study provides useful information on the energetics of molecules, which is related to their molecular structure. Presently, there are very few literature reports on the molecular energetics of five-membered nitrogenous heterocycles derived from hydantoin. For example, El-Sayed [16] reported the enthalpies of combustion and formation of hydantoin; Silva et al. [17] described a comprehensive computational and experimental energy study on hydantoin and 2-thiohydantoin. Recently, our research group has carried out thermochemical studies on the enthalpies of sublimation and enthalpies of formation in the condensed and gas phase of 5-methylhydantoin, 5-methyl-5-phenylhydantoin, and 5,5-diphenylhydantoin [18,19]. Nogueira et al. [20] conducted a study on the spectroscopic characterization and thermal behavior of 1-methylhydantoin. Among the reports mentioned, there is not one concerning to the thermodynamic properties of N-substituted hydantoin compounds. Based on the aforementioned, in this work we present a thermochemical study of 1-methylhydantoin (Figure 1) that was carried out, in order to contribute to its thermochemical characterization. The present work also includes a comparison between the energetic effect produced by a methyl in position 1 and that produced in other hydantoins by the same group in position 5. In this experimental study, thermochemical techniques such as differential scanning calorimetry, combustion calorimetry, and thermogravimetry, were used.

Static bomb combustion experiments of the interest compound were carried out, from which the specific combustion energy was determined. From this, the standard molar combustion energy and enthalpy and the standard molar formation enthalpy in the crystalline phase were calculated at *T* = 298.15 K. By thermogravimetric analysis, the mass loss rate of the liquid phase was measured as a function of the temperature and, applying the Langmuir equation, the enthalpy of vaporization was determined, as an intermediate point, to obtain the respective enthalpy of sublimation.

From the value of the enthalpy of formation in the condensed phase and the value of the enthalpy of sublimation, the standard molar enthalpy of formation in the gas phase was derived at *T* = 298.15 K for the compound studied. Based on the results obtained and with the information previously reported in the literature, an analysis of the structure-energy relationship with different hydantoin substituents was conducted.

## 2. Experimental Procedure

### 2.1. Material

1-methylhydantoin (CAS number: 616-04-6) was a commercial product purchased from Sigma-Aldrich (Merck KGaA, Darmstadt, Germany) with a purity of 97%. The compound was purified by sublimation under reduced pressure. After purification, the purity of the compound was determined by differential scanning calorimetry (DSC) using the fractional melting technique [21,22,23]. Table 1 summarizes the chemical data, supplier, and purity of the compound under study.

### 2.2. Physicochemical Properties by DSC

The purity and fusion temperature of 1MH were determined by DSC, using the fractional melting method [21,22,23]. The enthalpy of fusion was calculated as the area under the curve of the melting signal. For this purpose, a DSC Q2000 TA Instruments was used, which was previously calibrated in energy and temperature by means of metallic indium fusion experiments as reference material [25].

The first temperature sweep was carried out from room temperature to 450 K with a heating rate of 10 K·min^−1^. In this process, no impurities, crystalline transitions, or other thermal phenomena in addition to melting were detected. In the following experiments, heating was conducted in the temperature range of (420.15–440.15) K with a heating rate of 1 K·min^−1^ under a nitrogen flow of 50 cm^3^·min^−1^. Sample masses of approximately (1–3) mg were used in the experiments, which were placed in hermetically sealed aluminum cells. The masses of the samples for these experiments and those of fusion were weighed on a Mettler Toledo UMX2 balance (precision of 0.1 µg).

The description of the technique, as well as its practical applicability, are widely described in our previous work [19]. Table 2 compiles the physicochemical properties obtained by DSC from four experiments, where the uncertainty corresponds to the expanded one, which was calculated considering a *t*-student distribution with a coverage factor of *k* = 2.45 and a confidence level of 0.95. The details of all DSC experiments are shown in Appendix A. Additionally, the fusion entropy was calculated from the enthalpy and fusion temperature. Table 2 also shows some fusion properties for the compound studied, reported in other work taken from the literature. From comparison, a good concordance between them is observed.

### 2.3. Combustion Calorimetry

Combustion experiments were carried out in a static bomb isoperibolic calorimeter. This equipment has a Parr 1108 combustion bomb, made of stainless steel with an internal volume of 0.345 dm^3^. Additional details about the equipment and the technique used have been described in previous work [26,27].

Before carrying out the 1MH combustion experiments, the calorimeter was calibrated from combustion experiments of standard benzoic acid (NIST Reference Standard Material 39j), from which an energy equivalent of ε (calor) = (10123.0 ± 0.7) J·K^−1^ was obtained. The uncertainty corresponds to the standard deviation of the mean [27].

Once the calorimeter was calibrated, 1MH was burned in a tablet form and paraffin oil (Fluka, spectroscopic grade) was used as auxiliary material in order to obtain complete combustions. The value of the specific combustion energy (Δ_c_*u*°) of the paraffin oil is—(46.2356 ± 0.0026) kJ·g^−1^ with a molar mass and density of 14.02658 g·mol^−1^ and 0.857 g·cm^−3^, respectively [26].

Inside the combustion bomb, 1.0 cm^3^ of deionized water, the platinum crucible (containing the compound and the auxiliary material) was placed. A platinum wire, 0.05 mm in diameter and 50 mm long, closes the electrical circuit in the bomb head and the connection between the platinum wire and the sample was made by means of a cotton yarn (empirical formula: CH_1.742_O_0.921_ [26], combustion energy of Δ_c_*u*° = −(16.9541 ± 0.0031) kJ·g^−1^ [27]). The masses of the samples used in the combustion experiments were measured with a Sartorius ME 215S balance (precision ± 0.01 mg).

The combustion bomb was purged by passing high purity oxygen (Airgas Corp., x = 0.99999, Naugatuck, CT, USA) for five minutes and then filled to a pressure of 3.04 MPa. Subsequently, it was placed inside a brass bucket to which 2.0 kg of distilled water were added. The amount of water was measured with a Sartorius BP 12000-S balance (accuracy ± 0.1 g).

To start the combustion reaction, 4.184 J were supplied by a Parr 2901 ignition unit. The temperature recording during the combustion experiments was conducted with a Hart Scientific 5610 thermistor calibrated in the temperature range of (273.15–373.15) K. Resistance data were measured with an HP 34420A digital multimeter (sensitivity 10^−6^ kΩ). The resistance values obtained were converted to temperature values by means of a calibration equation.

The amount of nitric acid formed after combustion was recovered together with the bomb wash solution for subsequent titration using 0.01 M standard NaOH solution. The corrected temperature increase, the mass correction, and the Washburn corrections were obtained with computer programs developed in our laboratory, based on the information from the literature [28,29,30].

### 2.4. Thermogravimetric Analysis

The rate of mass loss was measured on a TGA Q500 TA Instruments, the two-arm balance of this equipment has an accuracy of 0.1 µg and was previously calibrated using NIST certified masses of 100 and 1000 mg. The temperature measurement was performed with a type R thermocouple that has a sensitivity of 0.1 K, which was calibrated by measuring the Curie temperature of alumel and nickel.

The mass loss rate of 1MH, in the liquid phase, was measured as a temperature function in the range of (440–460) K. The data obtained were plotted using Equation (1), which produces a straight line (ln (d*m*⁄d*t* · *T*) vs. 1⁄*T*) from whose slope the enthalpy of vaporization was obtained.
(1)$$\ln\left(dm/dt\cdot \;T\right)=B\;-\;\Delta_{l}^{g}H_{m}/RT\;$$

Equation (1) results from the combination of the Langmuir and Clausius-Clapeyron equations, where d*m*/d*t* is the mass loss rate at temperature *T*, $\Delta_{l}^{g}H_{m}$ is the enthalpy of vaporization, *B* is a constant involving other constants of the Langmuir and Clausius-Clapeyron equations, and *R* is the gas molar constant.

In each experiment, a sample (10–15) mg of 1MH was used, which was subjected to a heating of 10 K·min^−1^ from room temperature to 500 K with a gas nitrogen flow of 100 K·min^−1^. The usefulness and practicality of this technique to calculate enthalpies of vaporization was reported in a previous work [19].

## 3. Results and Discussion

Table 3 presents the results of seven combustion experiments of 1-methylhydantoin, as well as the average value of the mass combustion energy and the associated uncertainty, which corresponds to the standard one. These values refer to the idealized combustion reaction expressed in Equation (2) at *T* = 298.15 K. The details of all the experiments are shown in Appendix A.
(2)$$C_{4}H_{6}O_{2}N_{2}\left(\mathrm{cr}\right)+4.5O_{2}\left(g\right)=4{\mathrm{CO}}_{2}\left(g\right)+3H_{2}O\left(L\right)+N_{2}\left(g\right)\;$$

The standard molar combustion energy and enthalpy values, Δ_c_*U*° (cr) and Δ_c_*H*° (cr), and the standard molar enthalpy of formation, Δ_f_*H*° (cr), in the condensed phase at *T* = 298.15 K are presented in Table 4. According to the usual thermochemical practice, the uncertainties associated with these values correspond to the expanded ones with a confidence level of 0.95. Expanded uncertainty includes the uncertainty of the specific combustion energy of the compound, uncertainty of the calibration experiments and uncertainty of auxiliary materials [31,32]. To calculate the values of Δ_f_*H*° (cr), the standard molar enthalpies of formation at *T* = 298.15 K of H_2_O (L) and CO_2_ (g) were used, whose values are −(285.830 ± 0.042) kJ·mol^−1^ and −(393.51 ± 0.13) kJ·mol^−1^, respectively [33].

Uncertainties correspond to the expanded ones for a confidence level of 0.95 and include the contributions of calorimeter calibration and the energy of combustion of auxiliary materials.

Table 5 shows an experimental thermogravimetric series of the four that were carried out, this series includes the mass *m* that was measured at each temperature *T*, the mass loss rate d*m*/d*t*, and 1/*T*.

For each series, an adjusted linear equation and the correlation coefficient *r*^2^ were obtained. From the slope, the enthalpy of vaporization was calculated, which was associated with the experimental mean temperature <*T*_vap_>. From the linear fit, the uncertainties of the intercept *σ*_a_ and of the slope *σ*_b_ were calculated. The uncertainty of each vaporization enthalpy value corresponds to the combined uncertainty, *u*_comb_, which includes the uncertainties of the slope, temperature, and mass loss rate. From four vaporization enthalpy data, the weighted average and its respective uncertainty were calculated, which correspond to the standard uncertainty. The complete data are shown in Appendix A.

From the thermogravimetric data shown in Table 5, the enthalpy of sublimation was obtained at *T* = 298.15 K using the equations proposed by Chickos et al. [34], as described below: the enthalpy of vaporization at <*T*_vap_> was adjusted to *T*_fus_ = 431.0 K using Equation (3). At this temperature, the enthalpy of sublimation was calculated according to Equation (4) and used the enthalpy of fusion of the compound obtained by DSC in this same work. Finally, using Equation (5) the enthalpy of sublimation was obtained at *T* = 298.15 K. Where the $\Delta_{\mathrm{cr}}^{g}C_{p,\;m}^{o}$ was calculated with Equation (6) as recommended by Chickos et al. [35], and the $C_{p,\;m}^{o}\left(\mathrm{cr}\right)$ was considered as 107.01 ± 1.23 J·mol^−1^·K^−1^ [19]. The values of vaporization and sublimation enthalpies are shown in Table 6.
(3)$$\Delta_{l}^{g}H_{m}\left(T_{\mathrm{fus}}\right)=\Delta_{l}^{g}H_{m}\left(T_{\mathrm{vap}}\right)-\left[-0.0642\cdot \left(T_{\mathrm{vap}}-T_{\mathrm{fus}}\right)\right]$$
(4)$$\Delta_{\mathrm{cr}}^{g}H_{m}\left(T_{\mathrm{fus}}\right)=\Delta_{s}^{l}H_{m}\left(T_{\mathrm{fus}}\right)+\Delta_{l}^{g}H_{m}\left(T_{\mathrm{fus}}\right)$$
(5)$$\Delta_{\mathrm{cr}}^{g}H_{m}\left(298.15\;K\right)=\Delta_{\mathrm{cr}}^{g}H_{m}\left(T_{\mathrm{fus}}\right)-\left[\Delta_{\mathrm{cr}}^{g}C_{p,\;m}^{o}\cdot \left(T_{\mathrm{fus}}-298.15\;K\right)\right]$$
(6)$$\Delta_{\mathrm{cr}}^{g}C_{p,\;m}^{o}/J\cdot K^{-1}\cdot {\mathrm{mol}}^{-1}=-\left\{0.75+0.15\left(C_{p,\;m}^{o}\left(\mathrm{cr}\right)/J\cdot K^{-1}\cdot {\mathrm{mol}}^{-1}\right)\right\}$$

The enthalpy of sublimation previously reported in reference [19], which was obtained by thermogravimetry in our laboratory by measuring solid phase mass loss rates, was recalculated using Equations (5) and (6), and the result obtained was (94.6 ± 3.1 kJ·mol^−1^). Averaging the values, the one obtained in this work and the one taken from the reference, it can be considered that the enthalpy of sublimation of 1-methilhydantoin is 93.7 ± 2.2 kJ·mol^−1^, where the uncertainty was calculated as the average of the uncertainties. The final values obtained in present work are shown in Table 7.

In order to understand the influence of the methyl group on the 1-methylhydantoin, some gas phase enthalpies of formation of hydantoin and its derivatives substituted with phenyl and/or methyl groups were consulted (summarized in Table 8). From this table, the enthalpic change due to the presence of these substituents in hydantoin was calculated, as shown in Figure 2. It is seen that, the methyl group in hydantoin produces a stabilizing effect in both positions 1 and 5; however, the effect is greater at position 5 by 23.8 ± 5.3 kJ·mol^−1^. This is due to the change in the chemical environment of the methyl group. In 5-methylhydantoin, the methyl group is bonded to a carbon atom, which is less electronegative than the nitrogen atom at position 1. Also, methyl substitution at the 1-position is less exothermic than at the 5-position, presumably because of the loss of some intramolecular hydrogen bonding.

Contrary to this observation, when a phenyl group is inserted at the 5-position of 5-methylhydantoin, it tends to increase the enthalpy of formation by 143.6 ± 6.8 kJ·mol^−1^, and by 173.5 ± 5.6 kJ·mol^−1^ when a phenyl group replaces the methyl group in 5-methyl-5-phenylhydantoin, and both have a destabilizing effect. This behavior, as expected, is due to the electro-donating nature of methyl group, which tends to stabilize hydantoin, the opposite effect to phenyl, which is destabilizing.

## Figures and Tables

**Figure 1 molecules-27-00556-f001:**
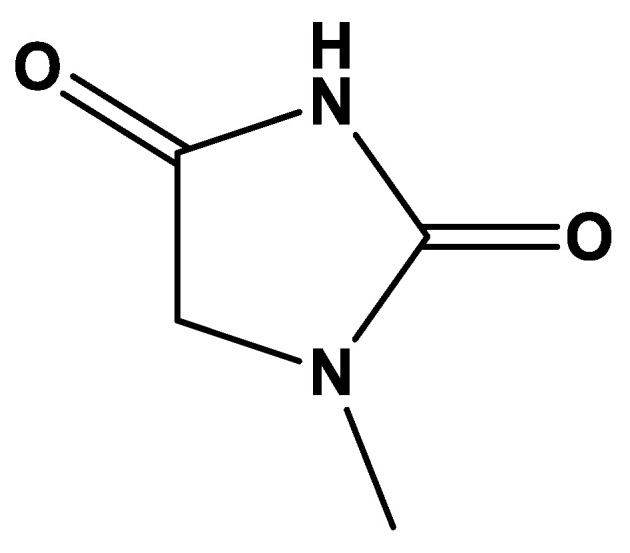
Molecular structure of 1-methylhydantoin (1MH).

**Figure 2 molecules-27-00556-f002:**
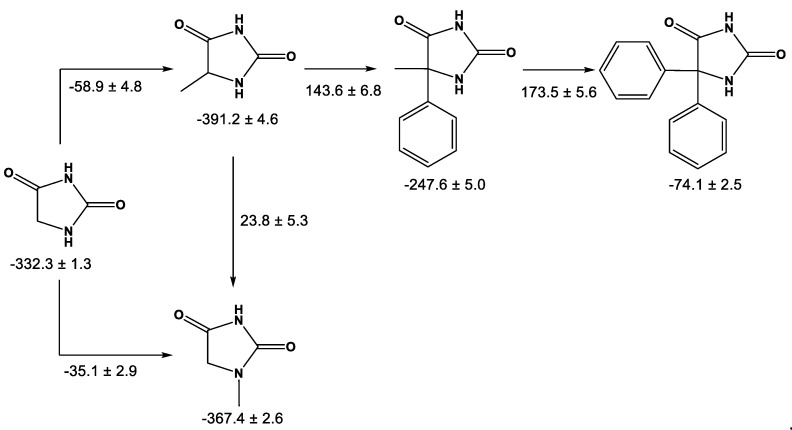
Comparative diagram of gas-phase enthalpies of formation of the 5 imidazolidine derivatives: 5,5-diphenylhydantoin, 5-methyl-5-phenylhydantoin, hydantoin, 5-methylhydantoin, and 1-methylhydantoin (1MH) studied in this work. All values in kJ·mol^−1^.

**Table 1 molecules-27-00556-t001:** Chemical data, source, and purities of the 1-methylhydantoin utilized in this work.

CAS Number	Source	$\frac{M{\;}^{a}}{g\cdot {mol}^{-1}}$	Initial Mole Fraction Purity ^b^	Purification Method	Final Mole Fraction Purity ^c^	Analysis Method
616-04-6	Aldrich	114.103	0.97	Sublimation	0.9995 ± 0.0002	DSC

^a^ The relative atomic mass was calculated followed the recommendations of the 2013 IUPAC commission [24]. ^b^ Values stated in the certificate of analysis provided by the supplier. ^c^ Results obtained from differential scanning calorimetry (the uncertainty quoted corresponds to the expanded uncertainty with a coverage factor of *k* = 2.45 and 0.95 confidence level for a *t*-student distribution).

**Table 2 molecules-27-00556-t002:** Physicochemical properties of 1-methylhydantoin at *P*° = 0.1 MPa obtained by DSC in this work and others reported in the literature.

Compound	$\frac{x}{\mathrm{mol}\;\mathrm{Fraction}}$	$\frac{T_{\mathrm{fus}}}{K}$	$\frac{\Delta_{\mathrm{cr}}^{l}H_{m}^{}}{kJ\cdot {mol}^{-1}}$	${\frac{\Delta_{\mathrm{cr}}^{l}S_{m}{(T}_{\mathrm{fus}})}{{J\cdot \mathrm{mol}}^{-1}{\cdot K}^{-1}}}^{b}\;$	Ref
1MH	0.9995 ± 0.0002 ^a^	431.0 ± 0.5 ^a^	22.02 ± 1.11 ^a^	51.1 ± 2.6	This work
	0.9996 ± 0.0001	430.9 ± 0.1	22.30 ± 0.11	51.8 ± 0.3	[19]
	-	428.9 ± 0.7	21.5 ± 0.3	50.1 ± 0.7	[20]

Standard uncertainty *u* is *u*(*P*) = 1 kPa. ^a^ The uncertainty corresponds to the expanded one, which was calculated considering a *t*-student distribution with a coverage factor of *k* = 2.45 and a confidence level of 0.95. ^b^ Uncertainty was calculated as $uu=\sqrt{\Delta_{s}^{l}S^{2}\left({\left(\frac{u_{\Delta_{\mathrm{fus}}H}}{\Delta_{\mathrm{fus}}H}\right)}^{2}+{\left(\frac{u_{T_{\mathrm{fus}}}}{T_{\mathrm{fus}}}\right)}^{2}\right)}.$

**Table 3 molecules-27-00556-t003:** Specific combustion energies of 1MH at *T* = 298.15 K and *p*° = 0.1 MPa.

−Δ_c_*u*°/(kJ∙g^−1^)
17.2798
17.2880
17.2810
17.2821
17.2727
17.2881
17.2657
〈−Δ_c_*u*°〉/(kJ∙g^−1^) = 17.2796 ± 0.0031

The uncertainty attached to the average of specific combustion energy is the standard deviation of mean, i.e., it is standard uncertainty.

**Table 4 molecules-27-00556-t004:** Standard molar properties of the 1-methylhydantoin, obtained from combustion calorimetry experiments, at *p*° = 0.1 MPa and *T* = 298.15 K.

$\frac{-\Delta_{c}U_{m}^{{}^{\circ}}\left(cr\right)}{kJ\cdot mol^{-1}}$	$\frac{-\Delta_{c}H_{m}^{{}^{\circ}}\left(cr\right)}{kJ\cdot mol^{-1}}$	$\frac{-\Delta_{f}H_{m}^{{}^{\circ}}\left(cr\right)}{kJ\cdot mol^{-1}}$
1971.65 ± 0.88	1970.41 ± 0.88	461.12 ± 1.37

**Table 5 molecules-27-00556-t005:** Thermogravimetric data of an experimental series and the vaporization enthalpy of 1MH.

$\frac{T}{K}$	$\frac{m}{\mathrm{mg}}$	$\frac{\left(dm/dt\right)\cdot 10^{9}}{\mathrm{kg}\cdot s^{-1}}$	$\frac{\left(1/T\right)\cdot 10^{3}}{K^{-1}}$	ln(d*m*/d*t* ⋯ *T*)
**Series 1**				
440.0	15.9006	0.0149	2.273	-18.846
442.0	15.7015	0.0161	2.262	−18.759
444.0	15.4900	0.0174	2.252	−18.678
446.0	15.2653	0.0188	2.242	−18.594
448.0	15.0273	0.0204	2.232	−18.512
450.0	14.7724	0.0220	2.222	−18.431
452.0	14.4989	0.0237	2.212	−18.351
454.0	14.2039	0.0255	2.203	−18.274
456.0	13.8841	0.0274	2.193	−18.197
458.0	13.5428	0.0295	2.183	−18.120
460.0	13.1747	0.0317	2.174	−18.043
Series 1 ln(dm/dt ·T) $=\;-0.4–8110.9/T;\;r^{2}\;=0.9999;\;\sigma_{a}\;=\;0.1;\;\sigma_{b}\;=17.9;\;\Delta_{l}^{g}H_{m}(450.0\;K)$/kJ·mol^−1^ = 67.4 ± 0.1*^a^*				
Series 2 ln(dm/dt ·T) $=\;-0.3–8156.2/T;\;r^{2}\;=\;0.9997;\;\sigma_{a}\;=\;0.1;\;\sigma_{b}\;=\;48.7;\;\Delta_{l}^{g}H_{m}(450.0\;K)$/kJ·mol^−1^ = 67.8 ± 0.4 *^a^*				
Series 3 ln(dm/dt ·T)$=\;-0.2–8201.4/T;\;r^{2}\;=\;0.9998;\;\sigma_{a}\;=\;0.1;\;\sigma_{b}\;=\;35.7;\;\Delta_{l}^{g}H_{m}(450.0\;K)$/kJ·mol^−1^ = 68.2 ± 0.3 *^a^*				
Series 4 ln(dm/dt ·T)$=\;-0.6–8032.0/T;\;r^{2}\;=\;0.9999;\;\sigma_{a}\;=\;0.1;\;\sigma_{b}\;=\;26.3;\;\Delta_{l}^{g}H_{m}(450.0\;K)$/kJ·mol^−1^ = 66.8 ± 0.2 *^a^*				
$\mathrm{Weighted}\;\mathrm{average}:\;<\Delta_{l}^{g}H_{m}$ (1MH, 450.0 K)>/kJ·mol^−1^ = 67.4 ± 0.1 *^b^*				

*^a^* The uncertainty corresponds to the combined standard one, which includes the uncertainties of the slope, the rate of mass loss and the temperature. *^b^* The uncertainty associated with the weighted average corresponds to the standard uncertainty.

**Table 6 molecules-27-00556-t006:** Vaporization and sublimation enthalpies of 1MH.

Compound	$\frac{{T_{\mathrm{vap}}}^{\;a}}{K}$	$\frac{\Delta_{l}^{g}H_{m}\left(T_{vap}\right){\;}^{b}}{kJ\cdot mol^{-1}}$	$\frac{\Delta_{cr}^{g}H_{m}\left(298.15\;K\right){\;}^{c}}{kJ\cdot mol^{-1}}$
1MH	450.0	67.4 ± 0.4	92.8 ± 1.2

*^a^* Standard uncertainty *u* (*T*) = 0.1 K. *^b^* The uncertainty corresponds to the expanded uncertainty with a coverage factor of *k* = 2.57 for a *t*-student distribution and a confidence level of 0.95. *^c^* The uncertainty was calculated by the method of the root of the sum of the squares of the uncertainties of the enthalpy of fusion and vaporization.

**Table 7 molecules-27-00556-t007:** Standard molar (*p*° = 0.1 MPa) enthalpies of sublimation, $\Delta_{\mathrm{cr}}^{g}H_{m}^{{}^{\circ}}$, and of formation in condensed phase $\Delta_{f}H_{m}^{{}^{\circ}}\left(\mathrm{cr}\right)$ and gas-phase, $\Delta_{f}H_{m}^{{}^{\circ}}\left(g\right)$, at *T* = 298.15 K for the 1-methylhydantoin.

${\frac{-\Delta_{f}H_{m}^{{}^{\circ}}\;\left(cr\right)\;}{kJ\cdot mol^{-1}}}^{a}\;$	${\frac{\Delta_{cr}^{g}H_{m}^{{}^{\circ}}\;}{kJ\cdot mol^{-1}}}^{a}\;$	${\frac{-\Delta_{f}H_{m}^{{}^{\circ}}\;\left(g\right)\;}{kJ\cdot mol^{-1}}}^{b}\;$
461.12 ± 1.37	93.7 ± 2.2	367.4 ± 2.6

*^a^* The uncertainties associated with each average value corresponds to the expanded uncertainty for a confidence level of 0.95. *^b^* Uncertainties calculated through the root sum square method.

**Table 8 molecules-27-00556-t008:** Enthalpies of formation, in gaseous phase, at *T* = 298.15 K of hydantoin and some of its derivatives.

Compound	${\frac{-\Delta_{f}H_{m}^{{}^{\circ}}\;\left(g\right)}{kJ\cdot mol^{-1}}}^{}$	Ref.
Hydantoin	332.3 ± 1.3	[33]
1-Methylhydantoin	367.4 ± 2.6	This work
5-Methylhydantoin	391.2 ± 4.6 ^a^	[19,35]
5-Methyl-5-phenylhydantoin	247.6 ± 5.0	[19]
5,5-diphenylhydantoin	74.1 ± 2.5	[19]

^a^ Value calculated from $\Delta_{f}H\left(\mathrm{cr}\right)=-486.6\;\pm \;1{.1\;\mathrm{kJ}\cdot \mathrm{mol}}^{-1}$ reported by Cox and Pilcher [36] and $\Delta_{\mathrm{cr}}^{g}H=95.4\;\pm \;4{.5\;\mathrm{kJ}\cdot \mathrm{mol}}^{-1}$ from reference [19].

## Data Availability

Not applicable.

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
