# Peer review of "Thermochemical Study of 1-Methylhydantoin"

_molecules, 2022, doi:10.3390/molecules27020556_

Round 1
Reviewer 1 Report
The article presents results from well-done experiment. But the scientific novelty is rather average.
Author Response
The authors welcome the reviewer's comments. Our research group has more than 20 years working in the experimental determination of thermochemical parameters of chemical systems of interest. We are convinced that these determinations are invaluable today, even for computational chemistry researchers, colleagues with whom we have worked closely.
Reviewer 2 Report
The authors report some thermochemical properties of N-methylhydantoin and discuss the relative stabilities of some methyl and phenyl substituted hydantoins. The experimental work seems satisfactory. There are some problems with the discussion regarding Figure 2 that needs to be addressed. I have edited the portion of the text dealing with Figure 2 assuming that the figure is correct. I don’t see what relevance the study of phenyl substitution on 5-methylhydantoin has on the discussion of phenyl substitution at the 5 position of 5-methyl 5-phenylhydantoin. In one case a hydrogen is replace and in the second, a methyl is replace, other than they are both endothermic. The comparisons are not equivalent. What is relevant to this study is that methyl substitution at the 1 position is less exothermic than at the 5-position presumably because of the loss of some hydrogen bonding. This is not mentioned.
I think the latter should be addressed by the authors.

Author Response
The authors appreciate the reviewer’s comments and suggestions. All of the suggested changes are included in the revised version.
About the discussion of phenyl groups substitution at the 5-position mentioned in the manuscript, is comparative with the effect caused by a methyl group. On the other hand, this mention allows us complementing the global work on substituted hydantoins that our research group has been carrying out in recent years, as indicated in references [18] and [19].
Reviewer 3 Report
I think this is a nice and well done work
Author Response
The authors appreciate the reviewer’s comments.
Reviewer 4 Report
See attached file

Author Response
The authors appreciate the reviewer’s comments and suggestions. All of the suggested changes were made in the text as recommended by the reviewer.
The enthalpy of sublimation at T=298.15 K reported in this work and that reported in reference [19] was recalculated using the Chickos equation suggested by the reviewer. The average sublimation enthalpy and the gas phase enthalpy of formation were recalculated from these new data.